Perception of color emotions for single colors in red-green defective observers

Sato Keiko satokei@eng.kagawa-u.ac.jp 1
Inoue Takaaki 2
1 Faculty of Engineering, Kagawa University , Takamatsu , Kagawa , Japan
2 Graduate School of Engineering, Kagawa University , Takamatsu , Kagawa , Japan
Abdullah Jafri
Electronic publication date: 2016 Dec 8
Publication date: 2016
Volume: 4
Electronic Location ID: e2751
Received 2016 Jun 14; Accepted 2016 Nov 3
Copyright: ©2016 Sato and Inoue
Copyright year: 2016
Copyright holder: Sato and Inoue
License: This is an open access article distributed under the terms of the Creative Commons Attribution License, which permits unrestricted use, distribution, reproduction and adaptation in any medium and for any purpose provided that it is properly attributed. For attribution, the original author(s), title, publication source (PeerJ) and either DOI or URL of the article must be cited.
License URL: https://creativecommons.org/licenses/by/4.0/

Keywords: Color emotion, Color vision, Color naming, Red-green color deficiency

Funding: CASIO Science Promotion Foundation JKA Promotion Foundation 27-177 This work was supported by the grants from CASIO Science Promotion Foundation and JKA Promotion Foundation (27-177), in Japan. The funders had no role in study design, data collection and analysis, decision to publish, or preparation of the manuscript.

==============================
It is estimated that inherited red-green color deficiency, which involves both the protan and deutan deficiency types, is common in men. For red-green defective observers, some reddish colors appear desaturated and brownish, unlike those seen by normal observers. Despite its prevalence, few studies have investigated the effects that red-green color deficiency has on the psychological properties of colors (color emotions). The current study investigated the influence of red-green color deficiency on the following six color emotions: cleanliness, freshness, hardness, preference, warmth, and weight. Specifically, this study aimed to: (1) reveal differences between normal and red-green defective observers in rating patterns of six color emotions; (2) examine differences in color emotions related to the three cardinal channels in human color vision; and (3) explore relationships between color emotions and color naming behavior. Thirteen men and 10 women with normal vision and 13 men who were red-green defective performed both a color naming task and an emotion rating task with 32 colors from the Berkeley Color Project (BCP). Results revealed noticeable differences in the cleanliness and hardness ratings between the normal vision observers, particularly in women, and red-green defective observers, which appeared mainly for colors in the orange to cyan range, and in the preference and warmth ratings for colors with cyan and purple hues. Similarly, naming errors also mainly occurred in the cyan colors. A regression analysis that included the three cone-contrasts (i.e., red-green, blue-yellow, and luminance) as predictors significantly accounted for variability in color emotion ratings for the red-green defective observers as much as the normal individuals. Expressly, for warmth ratings, the weight of the red-green opponent channel was significantly lower in color defective observers than in normal participants. In addition, the analyses for individual warmth ratings in the red-green defective group revealed that luminance cone-contrast was a significant predictor in most red-green-defective individuals. Together, these results suggest that red-green defective observers tend to rely on the blue-yellow channel and luminance to compensate for the weak sensitivity of long- and medium-wavelength (L-M) cone-contrasts, when rating color warmth.

Introduction

There are three types of photoreceptors in the human retina, short (S), medium (M), and long (L) wavelength-sensitive cones. Inherited color vision defects are characterized by the absence of a contribution from one or more of the cone photopigments, resulting in complete or partial loss of sensitive function (Neitz & Neitz, 2011). The type of cone that is lost or replaced by an anomalous photopigment defines different defects; a complete loss of cone function is called dichromacy, while a partial loss is called anomalous trichromacy (Merbs & Nathans, 1992). The most common color vision deficiencies are the protan and deutan types. The protan type, which includes protanopia and protanomaly, is caused by the replacement of the normal L gene with the M pigment gene or with an anomalous pigment gene. In contrast, in the deutan type, the normal M gene is replaced with the L pigment gene (deuteranopia), or with an anomalous pigment gene (deuteranomaly) (Neitz & Neitz, 2011; Smith & Pokorny, 2003). Individuals characterized as protan or deutan types are called red-green color vision defects. Red-green color vision defects are classified into the following four types: protanope, deuteranope (red-green dichromat), protanomalous, or deuteranomalous (red-green anomalous trichromats) (Neitz & Neitz, 2011). The incidence of deuteranomaly is the highest of four types (Neitz, Neitz & Kainz, 1996; Sharpe et al., 1999). Deuteranomalous individuals have varying degrees of trichromatic color vision with great variation. This differentiation stems from the S pigments and two narrowly separated photopigments that absorb in the L wavelength region of the spectrum. The severity of the deuteranomalous defect depends on the degree of similarity among the residual photopigments (Neitz, Neitz & Kainz, 1996).

Thus far, much research has contributed to the understanding of color perception in individuals with color vision defects. For example, previous studies have simulated how color appears for red-green defective observers who are dichromats (Brettel, Viénot & Mollon, 1997) and anomalous trichromats (Webster, Juricevic & McDermott, 2010). For red-green defective observers, some reddish colors appear more de-saturated and brownish than they do for normal vision observers. For red-green dichromats, when trichromatic cone-signals to a pair of colors differ only in the excitation of the missing cone, the dichromatic cone-signals are identical. They confuse such color pairs, due to a lack of the red-green opponent mechanism that is based on the comparison of the L and M cone responses (Vienot et al., 1995; Lillo et al., 2014). Other researchers have focused on color naming behavior for red-green dichromats and reported that they can sometimes name colors as accurately as normal observers, despite their visual impairment (Scheibner & Boynton, 1968; Smith & Pokorny, 1977; Nagy & Boynton, 1979; Montag & Boynton, 1987; Montag, 1994; Nagy et al., 2014). These studies presented colors using diverse methods, including presenting small stimuli (Scheibner & Boynton, 1968; Nagy et al., 2014) and large stimuli (Smith & Pokorny, 1977; Nagy & Boynton, 1979), with short and long durations (Montag & Boynton, 1987; Montag, 1994), indicating that red-green dichromats have a greater ability to name colors than predicted, especially for larger stimuli. Moreover, a recent study that explored the mechanism of color naming ability in red-green dichromats suggested that color naming ability is supported by residual activity in red-green opponent mechanism, as well as by blue-yellow and achromatic mechanisms (Moreira et al., 2014). As for anomalous trichromats, it has been found that red-green color appearance differs from normal vision, but it is recovered to a large degree by postreceptoral processing that compensates for the anomalous photopigment (Boehm, MacLeod & Bosten, 2014).

Further study by Álvaro et al. (2015) investigated the color preference of red-green dichromats. The results suggested that dichromats have different preferences than that of those with normal color vision; dichromats rated yellow best, unlike normal individuals. The authors examined color preference ratings for red-green dichromats and those with normal vision according to the fundamental neural dimensions that underlie color coding in the human visual system, which are mainly L-M and S-(L+M) cone-contrasts. The L-M cone-contrasts span from red to green approximately, while the S-(L+M) contrasts span from yellow to blue approximately. The color preference of red-green dichromats using two cone-contrasts based on a trichromatic model was additionally modeled by Álvaro et al. (2015); this resulted in failure as predicted. They also investigated whether corrected variables calculated based on dichromats’ cone responses (including lightness and saturation perception) could predict color preferences. The results showed that blue-yellow mechanism activity (an estimation for dichromats’ saturation perception) could explain the high variance in dichromats’ color preference. Other recent studies on normal trichromats have also used this method of examining color preference based on cone-contrast (Hurlbert & Ling, 2007; Palmer & Schloss, 2010; Taylor, Clifford & Franklin, 2013). Color preference based on vision characteristics should focus on cone-contrasts to better elucidate this mechanism.

In addition to color preference, individuals can also express various affective meanings of color, that is, color emotions which are defined as feelings evoked by colors (Ou et al., 2004). For instance, reddish colors are commonly called warm colors, while blue or greenish blue are cool colors. As another example, white, yellow, and light blue feel lighter in weight than do dark colors (Wright, 1962). Many experimental psychologists and psychometricians have explored the affective meaning of color using the semantic differential (SD) method introduced by Osgood, Suci & Tannenbaum (1957), as well as factor analysis (Wright & Rainwater, 1962) and principal component analysis (Hogg, 1969). These studies have attempted to identify the affective meanings of color using adjectives, summarizing various meanings based on evaluation, activity, warmth, impact, and so on. Several studies have also revealed a relation between adjectives or extracted factors and perception of color hue, lightness, and saturation (chroma) (Ou et al., 2004; Xin et al., 2004; Gao & Xin, 2006). However, no study to date has examined color emotions in red-green defective observers. Previously, the author has explored the affective color meanings of stimuli from the International Commission on Illumination 1976 (L*, u*, v*) color space (CIE LUV) in a small sample of individuals and suggested that it is highly likely that there are differences between normal and red-green defective observers for the perceived warmth a color has (Sato, Takimoto & Mitsukura, 2015).

The current study aims to expand on this work and compare color emotions between normal and red-green defective observers, and to model emotion ratings by cone-contrasts, replicating the study by Álvaro et al. (2015), who used BCP colors, but excluded muted tones. Therefore, in this study we expanded on these previous findings and included muted samples. Specifically, the current study used 32 colors from the BCP, comprised of saturated, light, muted, and dark samples of eight hues. Participants who had normal vision or who were red-green defective rated six color emotions for each color sample, and also named each one using 11 common basic color terms (BCTs). Six bipolar adjectives representing color emotions based on the study by Ou et al. (2004) were used as follows: clean–dirty (cleanliness), fresh–stale (freshness), soft–hard (hardness), like–dislike (preference), cool–warm (warmth), and light–heavy (weight). The resultant ratings were compared across groups. Then, using linear regression, ratings were predicted based on the cone-contrast model, consisting of red-green, blue-yellow, and luminance contrasts, and considered in terms of regression weights. Additionally, color emotion ratings were explored with regard to color naming behavior in the red-green defective group.

Materials and Methods

Subjects

The participants, all Japanese, included 13 men and 10 women with normal vision, and 13 men who were red-green defective observers, including both the protan and deutan types. Age is a potential factor influencing color preference, especially between younger and older women (Hurlbert & Owen, 2003). Thus, participants were sampled within a range of 20–40 years as much as possible. The mean ages in the normal men, normal women, and red-green defective observers were 23.23 years (SD = 4.00), 26.50 years (SD = 6.70), and 23.92 years (SD = 7.01), respectively, within an age range of 18–43 years. All participants were tested using Ishihara pseudoisochromatic plates. In addition, red-green defective individuals underwent testing with the Anomaloscope (NEITZ OT-ll) and then were diagnosed with the type and severity of color vision deficiency according to their matching, identifying two protanopes, one deuteranope, and 10 deuteranomalous trichromats. Red-green defective participants included all types of defects and severity as listed above. Written informed consent was obtained from all participants on an experimental protocol that was approved by the Ethics Committee of Kagawa University (26-002).

Equipment and stimuli

A computer (DELL Optiplex 9020) controlled test instrument and a calibrated monitor (EIZO ColorEdge CX270 with a resolution of 1,900 × 1,200 pixels), were used. Stimuli included 32 colors from the BCP (Palmer & Schloss, 2010), composed of saturated, light, muted, and dark samples of eight hues: red (R), orange (O), yellow (Y), chartreuse (H), green (G), cyan (C), blue (B), and purple (P). These eight hues were made up of the four Hering primaries (Stockman & Brainard, 2010), R, G, B, and Y, as well as four well-balanced binary hues, O, H, C, and P. The four color sets (i.e., saturated, light, muted, and dark) were defined by cutting color space that differed in the saturation and lightness levels. The 32 color stimuli in the current experiment were simulated as closely as possible on the monitor, which was white calibrated as a reference (x = 0.31, y = 0.32, Y = 99.93 cd/m2). The chromaticity coordinates for the 32 color stimuli were measured with a luminance and color meter (KONICA MINOLTA CS-150) and are shown in Table S1. The experiment was carried out in the dark room of a laboratory, on a calibrated monitor, under D65 ceiling lighting with a vertical luminance on the display of approximately 350 lux. The color stimuli were randomly displayed as circular patches at a viewing distance of 50 cm (19.68 inches) and a visual angle of four degrees. A chin rest was used to fix the distance between the participant and the screen, and color stimuli were presented at the center of the monitor at the participant’s eye level.

Procedure

The participants completed two tasks, which included a color naming and an emotion rating task involving six subtasks for six emotions, with roughly 1-minute intervals between tasks. The experiment was replicated at least two hours after the first measurement, with a reversal of the task order. Color stimuli were presented individually on a gray background (x = 0.32, y = 0.32, Y = 19.07 cd/m2).

In the color naming task, participants selected one color name from the 11 BCTs using a computer mouse: red, yellow, green, blue, purple, brown, orange, pink, black, white, and gray. These BCTs have been used for color naming across different languages and geographical locations (Berlin & Kay, 1969; Uchikawa & Boynton, 1987), and even across different color vision types (Bonnardel, 2006; Lillo et al., 2001; Lillo et al., 2014). In the present experiment, the buttons for the 11 BCTs were presented in Japanese below the color stimuli: aka (red), midori (green), ki (yellow), ao (blue), daidai (orange), momo (pink), murasaki (purple), tya (brown), kuro (black), hai (gray), and shiro (white). Participants responded by clicking on the color name as rapidly as possible. The buttons representing the color names (with each size being 2.1 degrees horizontal and 1.8 degrees vertical) were presented in a horizontal line with the order mentioned above. The order and position of each button was not changed throughout the task.

During the emotion rating task, participants rated six emotions using the bipolar adjectives. Six adjective pairs were chosen from a previous study by Ou et al. (2004), and presented in Japanese as follows: kirei-kitanai (clean–dirty), shinsen-furukusai (fresh–stale), yawarakai-katai (soft–hard), suki-kirai (like–dislike), tumetai-atatakai (cool–warm), and karui-omoi (light–heavy). The rating task for each emotion was carried out as a separate subtask, and counterbalanced across participants. Participants were instructed to rate their emotion for each stimulus by moving the mouse cursor along a horizontal bar representing an emotional range from zero to ten (17.9 degrees horizontal and 1.6 degrees vertical). For instance, in the case of a suki-kirai (like–dislike) rating, zero and ten describe “totemo suki (like very much)” and “totemo kirai (dislike very much),” respectively. The default position was set to the center of the bar, indicating neutral.

In both tasks, before the color stimuli appeared, a 500-ms fixation cross (1.3 degrees diameter) was presented. In the color naming task, color stimuli were presented for 10-s. If there was no response within the 10-s duration, the next stimulus was presented. In the emotion rating task, participants performed without a time limit, but were asked to respond as quickly as possible.

Results

Reliability and correlation between six color emotion ratings

Initially, in order to confirm the reliability of each color emotion rating, correlation coefficients were calculated between the first and second ratings for the six color emotions within each group (see reliability in Table 1). Reliability for all emotion ratings was high and significant in all groups (all p < .001). In addition, a correlation matrix was calculated to quantify the relationships between the ratings of the six adjectives for all groups (see correlation matrix between six emotions in Table 1). In the trichromatic male and red-green defective groups, only the warmth ratings showed no significant correlation to any other emotions. On the other hand, in the female observers, the hardness ratings also did not reveal any significant correlations to the other emotions except for the weight ratings. The other correlations were all significant (all p < .01).

Table 1 Reliability and correlation between six color emotion ratings.

Reliabilities (correlation coefficient between the first and second ratings) for six emotions were all significant (p < .001). In a correlation between six color emotion ratings, shaded cells indicate that no statistically significance were detected. The others were all significant (p < .01).

Group	Emotion	Reliability	Correlation between six color emotion ratings	
			Fresh–stale	Soft–hard	Like–dislike	Cool–warm	Light–heavy	
Men	Clean–dirty	0.79	0.97	0.88	0.91	−0.16	0.90	
Fresh–stale	0.72		0.84	0.91	−0.25	0.85	
Soft–hard	0.68			0.70	−0.30	0.96	
Like–dislike	0.64				−0.07	0.74	
Cool–warm	0.75					−0.13	
Light–heavy	0.71						
Women	Clean–dirty	0.84	0.96	0.22	0.87	0.08	0.86	
Fresh–stale	0.79		0.07	0.84	0.01	0.77	
Soft–hard	0.63			0.20	−0.28	0.56	
Like–dislike	0.80				0.13	0.71	
Cool–warm	0.78					0.09	
Light–heavy	0.84						
Red-green defects	Clean–dirty	0.78	0.96	0.91	0.90	−0.10	0.92	
Fresh–stale	0.69		0.86	0.92	−0.15	0.86	
Soft–hard	0.71			0.75	−0.15	0.95	
Like–dislike	0.75				−0.19	0.81	
Cool–warm	0.75					−0.09	
Light–heavy	0.72						

Color emotion ratings for normal and red-green defective observers

Figures 1 and 2 show the average color emotion ratings within each group of the four sets of colors (i.e., saturated, light, muted, and dark) along all eight hues (i.e., R, O, Y, H, G, C, B, and P). For cleanliness (clean–dirty) ratings in the trichromatic individuals, including men and women, the saturated and light samples were assessed as cleaner, with the exception of saturated purple, compared with the muted and dark colors, which were assessed as dirtier, regardless of hue. For both men and women, dark orange was rated the dirtiest. The red-green defective observers’ ratings were more varied across the four color sets, specifically in the green to cyan range, than were those of the normal participants. For the red-green defects, dark red was identified as dirtier, while light blue was the cleanest.

Figure 1 Mean ratings of cleanliness, freshness, and hardness (error bar means SEM) for 32 BCP colors consisting of saturated, light, muted, and dark sets.

Row means eight hues. These ratings were averaged for male, female, and red-green color defective observers separately.

Figure 2 Mean ratings of preference, warmth, and weight (error bar means SEM) for 32 BCP colors consisting of saturated, light, muted, and dark sets.

Row means eight hues. These ratings were averaged for male, female, and red-green color defective observers separately.

In the freshness (fresh–stale) ratings for the female group, there was a clear difference between each of the four color sets in the range of red to yellow. Similar to the cleanliness ratings, normal individuals rated the saturated and light samples as fresher, with the exception of saturated purple, while the muted and dark samples were rated as staler. For the red-green defective observers, light blue was rated the freshest, which was the same as in the cleanliness ratings, while dark orange was the stalest.

As for hardness (soft–hard), the normal men and red-green defective observers rated the light color sets as all softer, while the dark color sets were rated as harder. On the other hand, the female observers rated the saturated and dark color samples as harder compared with the light and muted sets.

For the preference (like–dislike) ratings, in order to facilitate comparison with previous results, we converted the rating scores to the inverse scale (0 for like, 10 for dislike). Normal individuals liked the saturated and light color sets, with the exception of saturated purple. Conversely, the muted and dark sets, with hues from red to chartreuse, were disliked. For the red-green defective observers, the preference pattern did not fluctuate much along the hue axis. Saturated orange was the most liked, similar to the men with normal vision, and muted purple was the least liked.

For all groups, the warmth (cool–warm) rating patterns steadily decreased from warm to cool in the red to blue range, but purple increased in warmth. Male observers tended to rate red, orange, and yellow hues in decreasing warmth according to the following order: saturated (warmest), light, muted, and dark (coolest). In addition, the women’s ratings varied across four sets in green and cyan hues. The four color sets (i.e., saturated, light, muted, and dark) were not as clearly separated by warmth ratings in the red-green defective observers.

For weight (light–heavy) ratings, all groups seemed to assign weight based on the four color sets. The light colors were rated lighter, while the dark color sets were heavier.

Mixed model ANOVAs were used to assess differences between groups in the six emotion ratings. For each emotion rating, a mixed-model ANOVA with the four color sets (i.e., saturated, light, muted, and dark), eight hues (i.e., R, O, Y, H, G, C, B, P), and three groups (male, female, and red-green defective observers) was conducted. Since the primary aim of this study was to examine differences between groups in color emotion ratings, only significant interaction effects that involved group differences (i.e., color set × group, hues × group, or color set × hues × group) were evaluated and are discussed here. Therefore, main effects of color set and hues, as well as the color set × hue interaction are not discussed. In addition, observed differences between groups based on Bonferroni-corrected pairwise comparisons are shown in Table 2.

Table 2 Observerd differences between groups.

M, F, and R stand for males, females, and red-green defective observers, respectively.

Cleanliness	Freshness	Hardness	
Dark orange	M = F > R*	Saturated set	F < M < R*	Saturated red	F > M = R*	
Muted yellow	F > R*	Muted set	F > M = R*	Light red	F < M = R*	
Dark yellow	M = F > R*	Dark set	F > M = R*	Muted red	F < M = R*	
Muted chartreuse	F > M > R*			Saturated orange	F > M = R*	
Dark chartreuse	M = F > R*			Light orange	F < R*	
Dark green	F > R*			Saturated yellow	F > M*	
Saturated cyan	M = F <R*			Dark yellow	F < M*	
Dark cyan	F > R*			Saturated chartreuse	F > M = R*	
				Dark chartreuse	F < M = R*	
				Saturated cyan	F > M = R*	
Preference	Warmth	Weight	
Cyan	F > R*	Saturated cyan	F < M = R*	Light set	F < M = R*	
Saturated purple	F < M = R*	Purple	F > R*	Dark set	F > M > R*	
Notes.

* p < .05.

For cleanliness ratings, the analysis showed a significant three-way interaction between group, color set, and hue (F42,693=2.17,p<.001,ηG2=0.04). To examine the three-way interaction, post-hoc two-way ANOVAs using the mean ratings of each hue were conducted with set and group as factors. The results revealed a significant interaction of group and color set on orange, yellow, chartreuse, green, and cyan (orange: F4.71,77.74=5.69,p<.001,ηG2=0.18; yellow: F3.70,61.11=3.46,p<.05,ηG2=0.12; chartreuse: F4.33,71.47=3.85,p<.01,ηG2=0.12; green: F4.38,72.23=2.99,p<.05,ηG2=0.10; cyan: F4.67,77.05=7.19,p<.001,ηG2=0.16). Simple effects showed that there were group differences on the dark sets for orange, yellow, chartreuse, green, and cyan (orange: F2,33=10.09,p<.001,ηG2=0.38; yellow: F2,33=6.67,p<.01,ηG2=0.29; chartreuse: F2,33=11.25,p<.001,ηG2=0.41; green: F2,33=3.90,p<.05,ηG2=0.19; cyan: F2,33=3.73,p<.05,ηG2=0.18), on the muted sets for yellow, and chartreuse (yellow: F2,33=3.97,p<.05,ηG2=0.20; chartreuse: F2,33=9.18,p<.001,ηG2=0.36), and on the saturated sets for cyan (F2,33=8.56,p<.01,ηG2=0.34).

For freshness ratings, the interaction of group and color set was significant (F6,99=9.93,p<.001,ηG2=0.34). Post-hoc one-way ANOVAs with group as a factor were done on the mean ratings of the four color sets. These analyses showed that there were significant group differences on the saturated, muted, and dark sets (saturated: F2,33=13.12,p<.001,ηG2=0.16; muted: F2,33=6.94,p<.01,ηG2=0.08; dark: F2,33=7.24,p<.01,ηG2=0.14).

The ANOVA for the hardness ratings revealed the significant interaction between group, color set, and hue (F42,693=1.75,p<.01,ηG2=0.03). To examine the three-way interaction, post-hoc two-way ANOVAs using the mean ratings of each hue were conducted with set and group as factors. The results revealed a significant interaction of group and color set on red, orange, yellow, chartreuse, and cyan (red: F6,99=6.84,p<.001,ηG2=0.19; orange: F4.64,76.51=4.30,p<.01,ηG2=0.15; yellow: F3.95,65.24=3.71,p<.01,ηG2=0.14; chartreuse: F4.79,79.10=7.56,p<.001,ηG2=0.23; cyan: F4.24,69.98=3.10,p<.05,ηG2=0.08). Simple effects showed that there were group differences on the saturated color sets for red, orange, yellow, chartreuse, and cyan (red: F2,33=9.20,p<.001,ηG2=0.36; orange: F2,33=4.63,p<.05,ηG2=0.21; yellow: F2,33=3.84,p<.05,ηG2=0.19; chartreuse: F2,33=19.38,p<.001,ηG2=0.54; cyan: F2,33=4.24,p<.05,ηG2=0.20), on the light sets for red and orange (red: F2,33=5.30,p<.05,ηG2=0.24; orange: F2,33=4.04,p<.05,ηG2=0.20) on the muted sets for red (F2,33=5.47,p<.01,ηG2=0.25), and on the dark sets for yellow and chartreuse (yellow: F2,33=4.00,p<.05,ηG2=0.20; chartreuse: F2,33=6.75,p<.01,ηG2=0.29).

For the preference (like-dislike) ratings, there was a significant three-way interaction between group, color set, and hue (F42,693=1.70,p<.01,ηG2=0.03). The post-hoc two-way ANOVAs using the mean ratings of the eight hues were conducted with color sets and group as factors. These results showed that for the cyan color, there was a significant group difference (F2,33=5.30,p<.05,ηG2=0.12) and for the purple color, group and color set interacted significantly (F6,99=2.54,p<.05,ηG2=0.08). Simple effects revealed group differences on the saturated color only (F2,33=5.17,p<.05,ηG2=0.24).

The ANOVAs for the warmth (cool-warm) ratings also revealed a significant three-way interaction between group, color sets and hue (F42,693=1.44,p<.05,ηG2=0.03). The post-hoc two-way ANOVAs with color sets and group as factors were done on the mean ratings of the eight hues. These results revealed that for the cyan color, group and color set interacted (F4.07,67.20=2.29,p=.067,ηG2=0.07). Simple effects showed group differences on the saturated set (F2,33=5.57,p<.01,ηG2=0.25). Furthermore, there was a significant group difference for purple (F2,33=3.57,p<.05,ηG2=0.04).

For weight ratings, the interaction between group and color set was significant (F4.42,72.96=5.34,p<.001,ηG2=0.08). Post-hoc one-way ANOVAs showed that there was a significant group difference for the light and dark sets (light: F2,33=5.96,p<.01,ηG2=0.14; dark: F2,33=10.44,p<.001,ηG2=0.18).

Prediction of color emotions based on three cone-contrast values

Similar to previous studies (Hurlbert & Ling, 2007; Palmer & Schloss, 2010; Taylor, Clifford & Franklin, 2013; Álvaro et al., 2015), the six emotion ratings were examined in relation to the three cone-contrast mechanisms based on a trichromatic vision system. This mechanism represents pairs of mutually exclusive perceptual categories of red-green (L-M), blue-yellow (S-(L+M)), and luminance cone-contrasts (L+M). Hurlbert & Ling (2007) summarized color preference by weights on two cone-contrasts, and Álvaro et al. (2015) and Taylor, Clifford & Franklin (2013) examined color preferences in terms of two cone-contrasts, lightness, and saturation values. Palmer & Schloss (2010) also used the two cone-contrast values, lightness, and saturation as predictors. In the present study, the two dimensions consisting of L-M and S-(L+M) cone-contrasts and the luminance cone-contrast value (lightness) was included. Because color emotions such as hardness or weight ratings may have a strong relation to lightness (Ou et al., 2004; Gao & Xin, 2006), and lightness of color has an influence on most color emotions (Xin et al., 2004).

Three cone-contrast values were calculated as follows according to the procedure of Álvaro et al. (2015). Firstly, the CIE x, y, and Y values were translated to the amount of L, M, and S cone excitations using Smith and Pokorny’s cone fundamentals (Smith & Pokorny, 1975). Secondly ΔL, ΔM, and ΔS, representing differences between each stimulus and the background were calculated using the following equations: ΔL = (Ls − Lb)∕Lb, ΔM = (Ms − Mb)∕Mb, and ΔS = (Ss − Sb)∕Sb. The subscript ‘s’ in the equations stands for the stimulus, while the subscript ‘b’ stands for the background color. Lastly, three cone-contrast values were calculated according to the cone-contrast weights that had been defined by Eskew, Mclellan & Giulianini (1999). Figure 3 shows the three cone-contrast values for the four color sets and eight hues.

Figure 3 Three cone-contrast values of 32 BCP colors.

(A–C) show L-M (red-green contrast), S-(L+M) (blue-yellow contrast), and L+M (luminance contrast), respectively.

The three cone-contrast values were used as predictors of the mean ratings for the six color emotions averaged within each group using linear regression. The resultant standardized regression coefficients, coefficients of determination (R squared), and F statistics are shown in Fig. 4. The prediction model that included the three cone-contrasts accounted for all emotion ratings significantly. Specifically, for the warmth and weight ratings, this model accounted for the high variances by more than 87% in all groups. The models for the cleanliness and hardness ratings also accounted for the high variances within a range of 65–85% with the exception of the hardness rating in the female group. For the freshness ratings, the model explained just half of the variances in the trichromatic group, while a higher variance was found in the red-green defective observers. Lastly, for the preference ratings, variances were lower, with the model accounting for about 40% in trichromatic observers, and 55% in the red-green defective group.

Figure 4 Regression weights (error bar means SEM) of the three cone-contrasts, coefficients of determination (R squared), and F statistics.

These figures illustrate the resulting regression weights in trichromatic males (light gray), trichromatic females (medium gray), and red-green defective observers (dark gray) for the six color emotion ratings.

In all of the regression models, except for the warmth ratings, the L+M contrast was a significant predictor of color emotion for all groups. The cleanliness model indicated that the S-(L+M) and L+M cone-contrasts were significant predictors of cleanliness ratings in all groups. In the freshness model, the S-(L+M) cone-contrast was a significant predictor of the freshness ratings in the trichromatic male and red-green defective group, but not in the female group. In the hardness model, the S-(L+M) cone-contrast was significant in the male and red-green defective groups. For the trichromatic male observers, the L-M cone-contrast was also a significant predictor. In the preference model, the L-M and S-(L+M) cone-contrasts were not significant predictors of preference ratings in any group. In the weight model, the S-(L+M) cone-contrast was a significant predictor of weight ratings for all groups. Lastly, for the warmth model in all groups, the L-M and S-(L+M) cone-contrasts were significant predictors of warmth ratings, but L+M was not significant. The regression weight of the red-green defective observers was much lower than those of trichromatic men or women.

Color naming results and the relation to color emotion ratings

Table 3 provides the color naming results of the normal including men and women and red-green defective groups. Red-green defective group included all types of defects and severity, following the evidence that protanope and deuteranope groups have similar color naming, with such similarities also apparent between protanomalous and deuteranomalous (Nagy et al., 2014). The naming responses and their frequencies are described in the table. If the naming responses of the red-green defective participants were not seen in the group with normal vision, they were underlined in the table. There were no missing response data in all participants.

Table 3 Naming responses and their frequencies for 32 BCP colors.

Naming responses with a frequency less than 15% were not shown in the table. If the naming responses of the red-green defective participants were not seen in the group with normal vision, they were underlined.

Color stimuli	Normals (%)	Red-green defects (%)	
Saturated red	Red(93)	Red(85)	
Light red	Pink(87)	Pink(77)	
Muted red	Red(57), pink(35)	Red(46), pink(31)	
Dark red	Red(87)	Red(50), brown(31)	
Saturated orange	Orange(100)	Orange(81), yellow(19)	
Light orange*	Orange(63), pink(28)	Pink(42), yellow(27), orange(27)	
Muted orange	Orange(65), brown(22)	Orange(35), brown(23), pink(19), red(15)	
Dark orange	Brown(96)	Brown(96)	
Saturated yellow	Yellow(98)	Yellow(100)	
Light yellow	Yellow(70), orange(24)	Yellow(73), orange(15)	
Muted yellow	Yellow(67), brown(20)	Yellow(35), green(19), orange(19), brown(19)	
Dark yellow*	Brown(72), yellow(17)	Brown(62), green(23)	
Saturated chartreuse	Yellow(76), green(24)	Yellow(92)	
Light chartreuse	Yellow(93)	Yellow(81), orange(15)	
Muted chartreuse	Green(50), yellow(48)	Green(65), yellow(15), brown(15)	
Dark chartreuse	Green(87)	Green(77), brown(23)	
Saturated green	Green(98)	Green(92)	
Light green	Green(76), blue(24)	Green(85)	
Muted green	Green(96)	Green(92)	
Dark green	Green(100)	Green(96)	
Saturated cyan***	Blue(89)	Green(50), blue(35), gray(15)	
Light cyan***	Blue(91)	Green(42), blue(38), white(15)	
Muted cyan***	Blue(83), green(17)	Green(73), gray(15)	
Dark cyan	Green(67), blue(33)	Green(81)	
Saturated blue	Blue(100)	Blue(92)	
Light blue	Blue(100)	Blue(85)	
Muted blue	Blue(100)	Blue(73)	
Dark blue	Blue(100)	Blue(54), green(19)	
Saturated purple	Purple(100)	Purple(81), blue(15)	
Light purple	Purple(74), pink(26)	Purple(65), pink(27)	
Muted purple	Purple(98)	Purple(81)	
Dark purple	Purple(98)	Purple(88)	
Notes.

* p < .05.

*** p < .001. p value adjustment by Holm.

To examine group differences in naming responses for the 32 colors, Fisher’s exact tests were conducted (2 groups × 11 BCTs). These results showed that naming responses were significantly different between groups for the light orange (p < .05), the dark yellow (p < .05), the saturated cyan (p < .001), the light cyan (p < .001), and the muted cyan (p < .001) colors.

Additionally, several variables were examined within each group in relation to performance on color naming, including response time, naming consistency, and naming consensus for each color stimulus, similar to what was done in the study by Álvaro et al. (2015). Naming consistency refers to the number of participants that used the same color name in the first and second responses. Naming consensus is the number of participants within a group that agreed with the most frequently used response. Correlation coefficients among these variables were calculated. There was a significant correlation between response time and consensus in all groups (men: r =  − 0.79, p < .001; women: r =  − 0.65, p < .001; red-green defects: r =  − 0.74, p < .001). There was no significant correlation between response time and consistency in all groups (men: r = 0.11, n.s.; women: r =  − 0.06, n.s.; red-green defects: r =  − 0.12, n.s.). Also, there was no significant correlation between consistency and consensus in all groups (men: r = 0.18, n.s.; women: r = 0.25, n.s.; red-green defects: r = 0.33, n.s.). Finally, correlation coefficients were calculated between the variables that influenced color naming performance and each emotion rating. As shown in Table 4, there were only significant correlations between the hardness ratings and response time, as well as between the hardness ratings and consensus, in the female group only.

Table 4 Correlation coefficients between six emotion ratings and response time, consistency, and consensus.

Group	Emotion	Response time	Consistency	Consensus	
Men	Clean–dirty	0.02	0.15	−0.06	
Fresh–stale	0.15	0.22	−0.14	
Soft–hard	−0.30	0.25	0.19	
Like–dislike	0.24	0.13	−0.25	
Cool–warm	0.25	−0.26	−0.34	
Light–heavy	−0.22	0.17	0.08	
Women	Clean–dirty	0.13	0.11	−0.14	
Fresh–stale	0.20	0.08	−0.20	
Soft–hard	−0.54**	0.16	0.39*	
Like–dislike	0.19	0.12	−0.17	
Cool–warm	0.31	0.08	−0.21	
Light–heavy	−0.15	0.10	0.12	
Red-green defects	Clean–dirty	−0.06	0.20	−0.19	
Fresh–stale	0.01	0.18	−0.23	
Soft–hard	−0.23	0.30	0.00	
Like–dislike	−0.01	0.15	−0.17	
Cool–warm	0.15	0.01	−0.03	
Light–heavy	−0.20	0.28	−0.09	
Notes.

* p < .05.

** p < .01.

Discussion

To date, much research has examined the fundamental emotional responses of human beings toward color. From the works, we know that colors can have affective meanings, and various factors influence the relationships between color emotions and color features such as hue, chroma, and lightness. In addition, there are strong universal trends in affective color meanings (Osgood & Adams, 1973). Despite the work that has been done investigating color emotions in individuals with normal vision, to date, there has been no work investigating this topic in color vision deficient observers, except for color preference (Álvaro et al., 2015). Therefore, the current study fills this gap in the literature by examining affective color meanings in red-green defective observers, and how the cone-contrast mechanism is related to perception of color emotions as a first approach.

The first significant finding from the current study was that color preference ratings were different in red-green defective observers compared to individuals with normal vision, similar to the findings of Álvaro et al. (2015). Álvaro et al. (2015) investigated color preference of red-green dichromats, as well as color naming behavior and the color vision mechanism, and showed different patterns in those with vision defects compared with those with normal vision; in dichromats, saturated yellow was liked the best, and protanopes’ preferences for cyan were lower than that of normal observers. This work used native Spanish participants and provided the typical color preference patterns of normal vision, and showed that normal observers like blue hues and dislike orange-yellow hues (see Fig. 1 in Álvaro et al. (2015)). In the American population, there was a higher preference for saturated colors, especially blue color was the most preferred, and dark yellow the least preferred (see Fig. 1 in Palmer & Schloss (2010)). Taylor, Clifford & Franklin (2013) compared the color preferences of Himba adults to those of British adults, and showed that Himba adults preferred the saturated colors for red, orange, yellow, chartreuse and green hues, but disliked bluish colors. Moreover, previous work that has investigated color preference of the Japanese (Saito, 1996) has found that white and black colors were liked, while dark purple, dark yellowish-brown, and dark red were disliked. Hurlbert & Ling (2007) have shown that a higher preference for reddish hues was seen in the Chinese population. It can be seen that, in color preference, there are not only underlying universalities but also cross-cultural variations. In the current study, both men and women with normal vision liked saturated and light colors with the range of green to blue, and dark yellow the least, similar to the previous studies. Interestingly, however, saturated orange and yellow were liked as much as green to blue hues. Additionally, compared with the preference patterns of normal observers obtained in Álvaro et al. (2015) or Palmer & Schloss (2010), it is clear that the normal participants rated color preference in order along the four color sets, especially in the range of orange to chartreuse hues. These discrepancies may come from cultural differences on color preference. Given that research says that the high preference for white and black colors was seen in the Japanese population, for example, Japanese might focus on lightness or saturation when rating color preference.

The preference rating patterns of the red-green defective group did not fluctuate across hues, unlike participants with normal vision. There was a significant difference between groups for the cyan and purple hues; the red-green defective observers preferred colors with a cyan hue less, a saturated purple more, compared to that of women, which is similar to the result of Álvaro et al. (2015). However, the significantly higher tendency to prefer yellow in red-green dichromats as seen in the study by Álvaro et al. (2015), was not indicated in the current study. A possible explanation for this difference is that red-green defective participants included not only dichromats, but also anomalous trichromats; 10 of 13 red-green defective observers were deuteranomalous trichromats. It is highly likely that the variability of severity of red-green deficiency influences color preference ratings. Figure S1 provides the emotion ratings for six weak deuteranomalous trichromats and four strong deuteranomalous trichromats, separately, to discuss the preference ratings of the red-green defective observers in more detail. The preference pattern of the participants with weak red-green defects was consistent without fluctuations across hues and color sets, but not among strong defects. There seemed to be differences in the preference ratings between the severity of red-green deficiency from weak to strong.

Similar to the preference ratings, there were also group differences in the other color emotions. For example, there were group differences in cleanliness ratings. It has been shown that ratings of color cleanliness are related to those of color preference, and both rating scales can be evaluated together as one factor (Osgood, Suci & Tannenbaum, 1957). In the current study, it was mainly found that female observers rated the dark orange, yellow, chartreuse, and green colors, and the muted yellow and chartreuse colors dirtier, and the saturated cyan cleaner than did the red-green defective observers. Furthermore, for the freshness rating pattern, group differences appeared in the saturated, muted, and dark color samples, regardless of hue. The normal participants rated the saturated samples fresher, and the women rated the muted and dark samples staler than did the red-green defective observers. As for the hardness rating, a sex difference was revealed in comparison with the other emotions. Female participants rated the saturated red, orange, chartreuse, and cyan colors harder, and the light and muted red and dark chartreuse colors softer, than did the normal men and the red-green defective observers. Moreover, with regard to the weight ratings, the normal and red-green defective observers both had similar patterns: light color sets were rated lighter and dark sets heavier, as expected. However, when comparing the groups, the red-green defective observers rated the dark colors heavier to a lesser degree than did the normal participants. In addition, the women rated the light set softer than did the normal men. Lastly, the warmth rating results showed that all color sets produced similar patterns along the hue axis. In studies that have tried to model color emotions on the basis of color features, it has been suggested that warm-cool rating is strongly connected to hue, and sometimes also chroma (Hogg, 1969; Ou et al., 2004; Xin et al., 2004). The differences in warmth rating between the normal, especially the women, and red-green defective observers were for the saturated cyan and purple colors. The female participants rated the saturated cyan cooler, and purple colors warmer, than did the red-green defective observers.

Regarding almost color emotion ratings, the results mainly showed rating differences between women and red-green defective observers, occasionally including men with normal vision. In general, women tended to rate color preference stronger than men, varying depending on the hue (Hurlbert & Owen, 2003). This tendency appeared in other color emotions as well as color preference. The observed rating differences between women and those in the red-green defective group may arise from a greater depth or diversity of color emotions in women. However, we need to state that these findings revealed from the current study containing sample size and distribution issues with red-green defective participants as already mentioned. As shown in Fig. S1, it can be seen that there are differences in emotion ratings, especially cleanliness or preference between weak and strong groups. It is likely that there are still many gaps in these findings. Further experiments should be performed to investigate color emotions of those in the red-green defective group not only by type but also degree to fill in these gaps in knowledge.

The second approach was performed to summarize color emotions in terms of cone-contrast mechanisms. Results from the study by Álvaro et al. (2015), which used the BCP colors without muted sets, showed that in normal trichromats (men and women), half or less of the variance in color preference was explained with S-(L+M) as the most significant predictor, but lightness was not. In contrast, for red-green dichromats, the trichromatic cone-contrast model could not predict color preference, due to the lack of L-M opponent response. Additionally, Álvaro et al. (2015) modeled dichromats’ color preferences using corrected cone-contrast values based on a dichromatic cone response. Their study indicated that the blue-yellow mechanism activity (an estimation for dichromats’ saturation perception) was a significant predictor, but corrected lightness was not. In the study by Palmer & Schloss (2010) for normal observers’ color preference, the S-(L+M) predictor accounted for the highest proportion of variance, and lightness and L-M predictors accounted for the lowest proportion of variance. Results from the study by Taylor, Clifford & Franklin (2013), which examined color preference in the British population, suggested that lightness was not a significant predictor, and that only L-M cone-contrast was a significant predictor in British males, and S-(L+M) cone-contrast in British females. In the current study, regression analyses were conducted for six emotion ratings using three cone-contrasts, namely L-M, S-(L+M), and L+M, which correspond to the red-green opponent activity, blue-yellow opponent activity, and luminance, respectively. In the normal group, both L-M and S-(L+M) were not significant predictors of color emotion preference, but L+M was. This result was not consistent with the previous studies listed above. This difference may arise because the normal participants tended to rate color preference in order along the four color sets, as already mentioned. The strong relationship to lightness can be clearly seen in color preferences (see preference rating patterns in Fig. 2), where preferences for the saturated and light sets appear stronger than those of the muted and dark sets.

Furthermore, in four of the emotion ratings, except for preference and warmth, the luminance contrast was a significant predictor in all groups. It is known that these four emotions are strongly related to lightness. Ou et al. (2004) modeled these ratings using CIELAB, a color space specified by the International Commission on Illumination that depends on lightness. The current study also suggests that these emotions are related to luminance in both the normal vision and red-green defective groups. The S-(L+M) cone-contrast was also significant predictors with the exception for the freshness and hardness rating in the women. The L-M cone-contrast was a significant predictor of the hardness ratings in the male group. For the warmth ratings, which are rated by hue, the L-M and S-(L+M) cone-contrasts, which involve two chromatic mechanisms, were significant predictors of warmth in the male and female groups with normal vision. The weight for the L-M cone-contrast in the red-green defective observers was also significant, and greatly lower than the normal group, as expected. In the current study, the red-green defective participants included both dichromats and anomalous trichromats; inclusion of both might have affected the results of the regression weights. Álvaro et al. (2015) found that the weight of the L-M cone-contrast was not a significant predictor for color preference in dichromats. To explore the difference in the weights for the L-M cone-contrast between dichromats and anomalous trichromats, the current study included a regression analysis that examined the three cone-contrasts as predictors of individual warmth ratings in the red-green defective group. Table 5 shows the weights of the three cone-contrasts and R squared. As mentioned above, the L-M cone-contrast was a significant predictor of the warmth ratings in the red-green defective group, which included both dichromats and anomalous trichromats. As shown in Table 5, for the three dichromats, including two protanopes and one deuteranope, it can be seen that the weight of the S-(L+M) cone-contrast was significantly higher than that of the L-M contrast, with the L-M cone-contrast identified as a significant predictor. Additionally, L+M contrast was a significant predictor. Surprisingly, for the three dichromats, the cone-contrast model accounted for more than half of the variance. Álvaro et al. (2015) indicated that deuteranopes might rate their color preferences using a residual red-green opponent activity for light colors. This suggestion is based on prior studies that argued dichromats use a residual red-green opponent activity, in addition to blue-yellow and achromatic mechanisms, when naming color stimuli (Moreira et al., 2014). The results for these individuals may also implicate the contribution of a residual red-green mechanism that influences dichromats’ ratings for color emotions. It also seems that the red-green defective observers rely on the L+M cone-contrasts in addition to the S-(L+M) to compensate for the weak sensitivity of the L-M cone-contrast when rating the warmth of color. In contrast, for anomalous trichromats with the deutan type of deficit, there seems to be quite a lot of difference in the weights between individuals. There are many variations of symptoms and severity in red-green deficiency; therefore, further research is necessary to explore how cone-contrast mechanisms affect color emotions in anomalous trichromats with different deficiency types and degrees.

Table 5 Regression weights of the three cone-contrasts and R squared for individuals warmth ratings in the red-green defective group.

Type and severity	L-M	S-(L+M)	L+M	R2	
Protanope	0.82**	−1.64***	−1.71***	0.74***	
Protanope	0.36*	−1.17***	−0.63***	0.78***	
Deuteranope	0.82*	−1.17**	−0.66†	0.52***	
Deuteranomalous trichromat (strong)	1.30***	−1.05***	−0.99***	0.78***	
Deuteranomalous trichromat (strong)	−0.09	0.02	0.67**	0.36**	
Deuteranomalous trichromat (strong)	0.71***	−1.24***	−0.12	0.84***	
Deuteranomalous trichromat (strong)	0.59*	−1.30***	0.94**	0.78***	
Deuteranomalous trichromat (weak)	1.11***	−1.43***	−0.95**	0.68***	
Deuteranomalous trichromat (weak)	0.52*	−0.94***	−0.17	0.60***	
Deuteranomalous trichromat (weak)	0.38*	0.07	1.09***	0.69***	
Deuteranomalous trichromat (weak)	1.34***	−0.95***	0.76***	0.90***	
Deuteranomalous trichromat (weak)	0.61***	−0.44**	−0.22†	0.71***	
Deuteranomalous trichromat (weak)	0.50*	−0.45	0.94***	0.60***	
Notes.

† p < .10.

* p < .05.

** p < .01.

*** p < .001.

The last aim of the study was to examine the relationship between color emotion ratings and color naming behaviors in the red-green defective observers. Although the only significant difference between groups was for the light orange, dark yellow, and cyan colors without the dark set, some of the red-green defective observers often responded with a different color name than the normal subjects for stimuli with low saturation, such as dark red, muted yellow, and dark blue. These naming behaviors might stem from the fact that dichromats make more naming errors for low saturation colors (Lillo et al., 2001). For the cyan colors, the majority of the normal group responded with the name blue. Conversely, more than half of the red-green defective observers named green except for the dark cyan set. In addition, around 15% responded gray or white. The participants who named achromatic colors such as gray or white for the cyan colors were protanopes. Normally, protanopes tend to make more errors than deuteranopes (Lillo et al., 2001). The other participants with deutan type, including deuteranope or deuteranomalous trichromats, responded with green more often than blue. The frequency with which red-green dichromats respond with green is much higher than in normal trichromats (Nagy et al., 2014). In addition, anomalous trichromats with the deutan type even respond with green more often than normal trichromats; they also tend to respond with green for the lightest blue and purple colors (Bonnardel, 2006). These results show that differences in the cleanliness or weight ratings between normal and red-green defective observers appear for dark color samples and saturated cyan. Therefore, it can be concluded that red-green defective observers rate color emotion differently than those with normal vision, especially for the color samples that they name in error. Although Álvaro et al. (2015) provided the evidence that the ease of color naming is related to color preference, in the current study, there was no significant relationship between response time and preference ratings. Participants in the study by Álvaro et al. (2015) were instructed to name verbally, while a mouse click response was used in the current experiment. This difference may influence the naming response time, as well as the arrangement of buttons representing color names.

Conclusion

Previous scientific studies have explored how emotional responses or preferences for colors can be affected by different theories, such as the ecological valence theory (Palmer & Schloss, 2010), the emotion-based theory (Ou et al., 2004), and the cone-contrast theory (Hurlbert & Ling, 2007). These findings were all obtained based on samples of normal trichromats with the exception of red-green dichromats’ color preferences (Álvaro et al., 2015). However, an individual’s color emotions come from experiences related to color during their life or the color vision mechanism that they have. Therefore, it is important to investigate how deviations in these experiences or mechanisms affect one’s color emotions. The current study specifically examined red-green defective observers’ preferences for colors and their emotional responses to them. These emotional responses were further explored in terms of a color vision mechanism and color naming behaviors. Based on the experimental results, the following conclusions can be made: (1) Red-green defective observers rate color emotions differently for some of dark colors in the orange to cyan range or the color cyan than do female observers with normal vision; (2) For the color cyan, red-green defective observers tend to make naming errors; and (3) Red-green defective observers likely rely on luminance and a blue-yellow opponent mechanism to compensate for the weak sensitivity of the red-green contrast when rating the warmth of colors.

In the current study, the red-green defective participants included both dichromats and anomalous trichromats. However, there are many variations of symptoms and severity in red-green deficiency, therefore, these findings revealed from the current study contain sample size and distribution issues with red-green defective participants. To reveal psychological properties of colors in red-green defective observers more profoundly, further research is necessary, using a large group with different types and severity of color deficiency and extending the study to achromatic color samples. However, it is highly likely that this study provides preliminary evidence of a mechanism that may be partially responsible for differences in color preferences and emotions in red-green deficient observers.

Supplemental Information

Figure S1 Mean ratings of 10 deuteranomalous trichromats for 32 BCP consisting of saturated, light, muted, and dark sets. Row means eight hues

These ratings were averaged for weak red-green defects group (N = 6) and strong defects group (N = 4), separately.

Click here for additional data file.

Table S1 CIE x, y, Y values of the 32 BCP color stimuli

Click here for additional data file.

Data S1 Raw data of color emotion rating and color naming

Click here for additional data file.

Additional Information and Declarations

Competing Interests

Author Contributions

Human Ethics

Data Availability

The authors declare there are no competing interests.

Keiko Sato conceived and designed the experiments, performed the experiments, analyzed the data, contributed reagents/materials/analysis tools, wrote the paper, prepared figures and/or tables, reviewed drafts of the paper.

Takaaki Inoue performed the experiments, contributed reagents/materials/analysis tools.

The following information was supplied relating to ethical approvals (i.e., approving body and any reference numbers):

The Ethics Committee of Kagawa University (26-002).

The following information was supplied regarding data availability:

The raw data has been supplied as a Supplemental File.

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
