# Peer review of "Perception of color emotions for single colors in red-green defective observers"

_PeerJ, doi:10.7717/peerj.2751_

## Round 0.1 · original submission · Major Revisions

· Academic Editor

Major Revisions

Dear Authors,

Please heed the comments of Reviewer 1 who has many important comments to improve the manuscript and the comments from Reviewer 2 which said: "It was interesting the see that the result was inconsistent with previous study, and the author speculate it might be due to the participant clinical condition (line 321-323). Is there a possibility an analysis be done in comparing between the dichromats and anomalous trichromats?"

Please also send the revised manuscript for English editing and resend the revised manuscript with a proof of English editing from a professional syntax and grammar editing company.

Reviewer 1 ·

Basic reporting

The submission adheres to all PeerJ policies.
However some parts of the paper are not written in a clear English or contain theoretical inaccuracies. I will describe some of them:
- Lines 32-33: “The type of cone that is lost or replaced by an anomalous photopigment defines different defects; a complete loss of cone function is called dichromats, while partial loss is called anomalous trichromats”. Different defects are dichromacy and anomalous trichromacy. Dichromats and anomalous trichromats are the labels for the individuals affected by dichromacy or anomalous trichromacy.
- Line 33: “The most common color deficiencies”. I recommend using “color vision deficiencies”.
- Lines 34-36: “In the deutan type, the L cone is normal but the M cone that is close to the normal L is completely or partially lost. In contrast, the protan type has a normal M cone and a lack of an L cone, or anomalous L cone function (Nathans et al., 1986)”. This definitions are not clear or accurate. Please review Neitz and Neitz (2011) and Smith and Pokorny (2003, Color matching and color discrimination. In S. K. Shevell (Ed.), The Science of Color (2nd ed., pp. 103-148). Oxford, UK: Optical Society of America.
- Line 39: “The most common classification is the deuteranomalous trichromat type (Neitz et al., 1996).” This sentence is not clear.
- Lines 49-51: “In particular, red-green dichromats confuse red-green hues due to a lack of the red-green opponent mechanism that is based on the comparison of the L and M cone responses (Vienot et al., 1995).”. This sentence is a wrong simplification of red-green dichromacy and its functional consequences. Red-green mechanism responds to a great variety of stimuli. The confusion between red and green hues is very unusual for surface or computerized stimuli – the results in table 2 are a good example.
- Lines 58-60: “Moreover, a recent study that explored the mechanism of color naming ability in red-green dichromats, suggested that color naming ability is supported by residual red-green cones, as well as yellow-blue and achromatic mechanisms (Moreira et al., 2014).”. It was really surprising to read about residual red-green cones. I checked the original paper and the authors do not propose the existence of residual red-green cones. They talk about red-green “residual activity” or “residual discrimination” and the different hypotheses for the existence of this residual mechanism, but “red-green residual cones” are not one of these hypotheses.
- Table two: please use deuteranomalous instead of deuteranomalos.

Experimental design

The experimental design presents some problems that must be solved prior to publication:
- Sample size and sample grouping: there exist a lot of individual differences in color preference data. In order to compensate for such differences, color preference studies require bigger sample sizes than the ones used in the current study. In relation to common observers, previous studies have reported differences between males and females in color preference (e.g. Hurlbert & Ling, 2007). It would be advisable to include more participants and to analyze the data separately for males and females. In relation to the red-green defective observers, the group is quite small and heterogeneous. From lines 307-314, it seems that previous literature (Alvaro et al., 2015) reports different color preferences for protanopes and deuteranopes. There are also some big differences between anomalous trichromats and dichromats in color perception. Therefore it seems sensible to analyze separately the data from protanopes, deuteranopes, protanomalous and deuteranomalous. The current sample size does not allow to do that so I recommend to increase the number of participants. Age is another potential factor of influence in color preference (see Hurlbert and Owen, 2015, In Elliot, A., Fairchild, M. & Franklin, A. (eds.) Handbook of Color Psychology, pp. 454-477. Cambridge: Cambridge University Press). It would be ideal to control also the age differences between the different groups if it is possible.
- The chromatic measurements have been made with an “EIZO Spyder4/EX2”. This unit performs the measurements attached to the screen so it is not optimal for measuring both the computer stimulation and the “D65 ceiling lighting” used during the experiment. If the author only has the “EIZO Spyder4/EX2” available, I recommend to do the experiment in a dark room. If there is other tool available, I recommend to measure the stimuli under the “D65 ceiling lighting”. In this case, it would be advisable to include the D65 intensity in lux units to allow reproducibility.
- As it is well-commented in the discussion, previous data from Saito (1996) found high preferences for black and white, so I would recommend, if possible, to extend the study to some achromatic samples.
- It would be also interesting to discuss the results in relation to previous results with other cultures (see Hurlbert and Owen, 2015, p. 464).

Validity of the findings

The statistical analyses performed seem reasonable and in agreement with previous research. The presentation of the results in the preference graph is not very intuitive. Previous literature has used the inverse scale (10 for like, 0 for dislike). If due to cultural reasons the Japanese population founds more intuitive to answer to the inverse scale (0 for like, 10 for dislike), I suggest to convert the results to the usual scale (10 for like, 0 for dislike) to facilitate comparisons with previous results.
The conclusions are not robust due to the sample problems previously described. After increasing the sample size, I recommend repeating the analyses including comparisons between different groups. The results section should be re-revised after that.

Additional comments

Overall I think this research has great interest for the scientific community and I want to value the effort of the author in conducting this research. However there are major issues to address in order to consider it for publication.

Reviewer 2 ·

Basic reporting

1. The author used clear and unambiguous professional English language throughout the article.
2. Relevant introduction and literature were provided
3. The structure is acceptable.
4. Figures and tablets are well described and labelled

Experimental design

Experimental Design
1. Research question is well addressed, relevant and meaningful.
2. The knowledge gap is clearly stated.
3. The experimental design is performed to ethical standard
4. Method is well described, an illustration on the visual stimuli and procedure might be useful in understanding the experiment.
5. Were the participants Japanese? (useful information fro cross-cultural study)

Validity of the findings

1. For the data analysis are statistically sound
2. It is interesting the see that the result was inconsistent with previous study, and the author speculate it might be due to the participant clinical condition (line 321-323). Is there a possibility an analysis be done in comparing between the dichromats and anomalous trichromats?

Additional comments

1. Conclusion is well stated and clear with the research question.
2. Possibly suggestion for future research might be important to be added.
3. In general, the research report is well prepared with sufficient data, appropriate analysis and conclusion.

---

## Round 0.2 · Major Revisions

· Academic Editor

Major Revisions

Dear Author,

Please re-revise the manuscript as per the comments of both peer reviewers especially the validity of the findings.

Thank You.

Reviewer 1 ·

Basic reporting

The submission adheres to all PeerJ policies.

Experimental design

The author has solved some weaknesses of the study. Other potential improvements were not easy to implement and the author has included them as future research lines. The author has done a valuable effort.

There are other few things to improve:

Line 78-79: “Alvaro et al. (2015) modeled color preference of red-green dichromats using a cone-contrasts model, which resulted in failure as predicted”. Cone-contrast model should be used. Correct along the text. Cone-contrast (singular) should be used when talking about the model/theory or about a single mechanism (i.e. L-M). Cone-contrasts (plural) must be used when talking about two or more mechanisms (i.e. Line 262 should say cone-contrast mechanisms, using plural). In addition, the referenced study found the cone-contrast model partially explaining red-green dichromat’s colour preference. From their abstract: “Trichromats’ preferences were summarized effectively in terms of cone-contrast between color and background, and yellow-blue cone-contrast could account for dichromats’ pattern of preference, with some evidence for residual red–green activity in deuteranopes’ preference.”

Text in lines 121-122 is not appropriate for the subjects section: “As in Nagy et al. (2014), protanope and deuteranope groups have similar color naming, with such similarities also apparent between protanomals and deuteranomals.”.

Lines 152-153: colour naming task is described but the screen arrangement of the buttons for the 11 BCTs is not clear. It must be explained because it might have influenced the response time variable and in such a case it must be discussed as a limitation in the discussion.

It would be advisable to include the Japanese words used for the bipolar adjectives to enhance reproducibility (lines 156-157).

Line 266 “However, in the present study, the luminance cone-contrast was also included, even though the previous study did not include it in the prediction model, because color emotions such as hardness or weight ratings may have a strong relation to lightness (Ou et al., 2004; Gao and Xin, 2006), and lightness of color has an influence on most color emotions (Xin et al., 2004).”. Lightness has been included as a predictor in previous colour preference studies (see Palmer and Schloss, 2010; Taylor, Clifford & Franklin, 2013; Alvaro et al, 2015).

The underlining criterion in Table 3 needs further clarification.

Line 336: “preferences for cyan was lower” must read “were lower”.

Figure 4 needs to clarify the meaning of the three grey bars (male, female and r-g anomalous?).

Line 460: genetic-based theory is not the usual term. It is advisable to use cone-contrast theory.

Line 461: “These findings were all obtained based on samples of normal trichromats.” This line needs to include the exception of the dichromats study (Alvaro et al.).

Validity of the findings

This study is valuable in measuring the colour emotions of anomalous observers, but the methodological weaknesses make advisable to take it as a first approach. Discussion does not take into account the limitations of the study in terms of sample size and sample distribution. For example, the flatness of the preference ratings in red-green anomalous observers may arise from the variability of the colour vision defects represented in figure 2. Even if we analyse separately the deuteranomalous trichromats (Figure S1), this group might include observers with different degrees of anomalous deuteranomaly (strong or weak defects) which results in very different colour perception. This is a strong limitation of the study and it needs to be clearly stated and taken into account through the discussion section.

Discussion needs to include information about the inclusion of lightness dimension in previous studies (Palmer and Schloss, 2010, Alvaro et al, 2015, Taylor, Clifford and Franklin, 2013).

Additional comments

The paper shows novel evidence for colour emotion theory and it can be considered a good first approach to the topic. However the discussion and conclusion sections need to be cautious because of the methodological limitations.

Reviewer 2 ·

Basic reporting

1. In general, the fundamental scientific report writing is sufficient, however English language proof reading is highly recommended. For instance, it is suggested to modify

Line 393: As mentioned already…
to ‘As already mentioned’ or ‘As mentioned earlier’ or ‘As previously stated’.

2. Proof reading is highly recommended.

Experimental design

1. The research question and objective is clearly stated in the text which contributed to filling the gap by using different sets of sample in color emotion study

2. For the method section, it is well described with sufficient information. However, It is suggested to add the information about the inter-trial interval (ITI) in the procedure section.

3. Style of writing can be further improved.

Validity of the findings

1. The results were reported with statistically sound and appropriate statistical procedure.

2. An appropriate conclusion has been made and consistent with the research question as stated.

3. The information about the effect size would be useful in the statistical report

4. Line 165-166 :...If there was no response within the 10-s duration, the next stimulus was presented…

How do you treat the no response cases? Is there any significant finding in terms of no response trials between normal and red-green defect group?

Additional comments

Overall, this article is well written with some minor improvement needed to be made, especially in the English language.

---

## Round 0.3 · Minor Revisions

· Academic Editor

Minor Revisions

Dear Author,

Please perform final minor editing of the manuscript according to the comments of the first Peer Reviewer.

Thanking you.

Reviewer 1 ·

Basic reporting

The submission adhere to all PeerJ policies.

Experimental design

The experimental design is robust and the limitations have been stated in the text.

Validity of the findings

The findings are robust and the limitations have been stated in the text.

Additional comments

Lines 78-81 needs rewording: They also investigated whether corrected variables calculated based on dichromats’ cone responses for lightness and saturation could predict color preferences. The results showed that saturation, which indicating blue-yellow mechanism activity, could explain the high variance in dichromats’ color preference.

The sentence “dichromats’ cone responses for lightness and saturation” is strange. I would suggest “dichromats’ cone responses (including lightness and saturation perception)”.

The sentence “The results showed that saturation, which indicating blue-yellow mechanism activity,” is strange. I would suggest “The results showed that blue-yellow mechanism activity (an estimation for dichromats’ saturation perception),”

Lines 161-162: It states that the adjectives were presented both in English and Japanese. It is strange since the participants’ section states that all of them were Japanese. Please clarify.

Line 168: “a 500 ms fixation cross (17.9 degrees diameter)”. Please confirm the size of the fixation
cross: it seems too big for stimulus of 4 degrees.

Caption of Figure 4 needs rewording: “These figures illustrate the resulting regression weights in each group for the six color emotion ratings. Three gray bars mean trichromat male, female, and red-green defective observers, respectively.”

I would suggest: “These figures illustrate the resulting regression weights in trichromatic males (light grey), trichromatic females (medium grey) and red-green defective observers (dark grey) for the six color emotion ratings.”

Line 317: for consistent nomenclature through the text, please use “protanomalous and deuteranomalous” instead of “protanomals and deuteranomals”.

Lines 426-429: The sentence “Additionally, Álvaro et al. (2015) modeled dichromats’ color preferences using corrected cone-contrast values based on a dichromatic cone response, which indicated that the corrected saturation, which relating to blue-yellow mechanism activity, was significant predictor, but corrected lightness was not.”

I would suggest: “Additionally, Álvaro et al. (2015) modeled dichromats’ color preferences using corrected cone-contrast values based on a dichromatic cone response. Their study indicated that the blue-yellow mechanism activity (an estimation for dichromats’ saturation perception) was a significant predictor, but corrected lightness was not.”


Lines 430-431: “the S-(L+M) predictor had the highest variance, and lightness and L-M predictors were the lowest”.

I would suggest “the S-(L+M) predictor accounted for the highest proportion of variance, and lightness and L-M predictors accounted for the lowest proportion of variance”

Line 433: “Britich males” should read “British males”.

Reviewer 2 ·

Basic reporting

The author have make an improvement in the article based on the commented points highlighted by the reviewer previously including in the method and results sections.

Experimental design

1. The experimental design have been well describe with sufficient information and the author manage to further improve by adding relevant points that have been raised by the reviewers.

Validity of the findings

no comment

Additional comments

In general, several important points and issues raised before have been well addressed by the author and a significant improvement have been made in the text, including the overall style of writing, language, the detail information of the study, result and discussion sections.

---

## Round 0.4 · accepted · Accept

· Academic Editor

Accept

Dear Authors, Thank you for the re-submission of this manuscript which has been accepted and will be processed for further proof preparation.

Reviewer 1 ·

Basic reporting

The submission adhere to all PeerJ policies

Experimental design

The submission discribes original primary research within the Scope of the journal

Validity of the findings

The findings are valid

Additional comments

No additional comments